# Distinct populations of crypt-associated fibroblasts act as signaling hubs to control colon homeostasis

**Michael David Brügger**[1⊕], **Tomas Valenta**[1,2⊕]*, **Hassan Fazilaty**[1⊕], **George Hausmann**[1], **Konrad Basler**[1]*

**1** Department of Molecular Life Sciences, University of Zurich, Switzerland, **2** Institute of Molecular Genetics of the Czech Academy of Sciences, Prague, Czech Republic

⊕ These authors contributed equally to this work.
* tomas.valenta@mls.uzh.ch (TV); konrad.basler@imls.uzh.ch (KB)

**Data Availability Statement:** Single-cell RNA sequencing data generated in this study are deposited in GEO with the accession number GSE151257. All data needed to reproduce the

## Abstract

Despite recent progress in recognizing the importance of mesenchymal cells for the homeostasis of the intestinal system, the current picture of how these cells communicate with the associated epithelial layer remains unclear. To describe the relevant cell populations in an unbiased manner, we carried out a single-cell transcriptome analysis of the adult murine colon, producing a high-quality atlas of matched colonic epithelium and mesenchyme. We identify two crypt-associated colonic fibroblast populations that are demarcated by different strengths of platelet-derived growth factor receptor A (Pdgfra) expression. Crypt-bottom fibroblasts (CBFs), close to the intestinal stem cells, express low levels of Pdgfra and secrete canonical Wnt ligands, Wnt potentiators, and bone morphogenetic protein (Bmp) inhibitors. Crypt-top fibroblasts (CTFs) exhibit high Pdgfra levels and secrete noncanonical Wnts and Bmp ligands. While the Pdgfra^low cells maintain intestinal stem cell proliferation, the Pdgfra^high cells induce differentiation of the epithelial cells. Our findings enhance our understanding of the crosstalk between various colonic epithelial cells and their associated mesenchymal signaling hubs along the crypt axis—placing differential Pdgfra expression levels in the spotlight of intestinal fibroblast identity.

## Introduction

The architecture of the colon is made up of diverse cell types, including undifferentiated and various differentiated epithelial cells whose fates are tightly programmed by signals from both the epithelium and the underlying mesenchyme. The intestinal mesenchyme displays great cellular heterogeneity, consisting, among others, of myofibroblasts, fibroblasts, smooth muscle, and endothelial cells [1]. It is an important player in intestinal homeostasis and partakes in the multitude of signaling pathways that are integral to the maintenance of epithelial stem cells, as well as their subsequent differentiation.

The exposure to a hostile environment and constant wear and tear requires the intestinal epithelium to replenish roughly every 5 to 7 days. The epithelial dynamics are driven by stem

figures in this publication can be found in the supporting information. The FACS sorting plots in Fig 1B, Fig 3D and S4C Fig are based on the .fcs data files in n S2 Data, S3 Data, S4 Data and S5 Data respectively. Transcript expression data for the violin plots in Fig 2C and 2D and Fig 4C and 4D can be found in S1A, S1B, S1E and S1F Data respectively. Fluorescence intensity measurements in Fig 3C and statistical analysis can be found in S1C Data. Raw data for the qPCR data in Fig 3E can be found in n S1D Data.

**Funding:** This work was supported by the Swiss National Science Foundation (http://www.snf.ch/de/Seiten/default.aspx), the University of Zurich Research Priority Program (URPP) "Translational Cancer Research" (https://www.cancer.uzh.ch/en.html) and the Kanton of Zürich (https://www.zh.ch/de.html). MDB is supported by the University of Zürich Forschungskredit Candoc grant Nr. FK-19-074 (https://www.research.uzh.ch/en/funding/phd/fkcandoc.html). TV is supported by Czech Science Foundation grant 18-21466S (https://gacr.cz/en/) and is a fellow of the URPP Translational Cancer Research (https://www.cancer.uzh.ch/en.html). HF is a fellow of the Cancer Research Center Zurich (CRC) (http://www.cancercenter.usz.ch/forschung/cancer-research-center/Seiten/default.aspx). The funders had no role in study design, data collection and analysis, decision to publish, or preparation of the manuscript.

**Competing interests:** The authors have declared that no competing interests exist.

**Abbreviations:** Bmp, bone morphogenetic protein; CBF, crypt-bottom fibroblast; CTF, crypt-top fibroblast; EGF, epidermal growth factor; ERK, extracellular signal-regulated kinase; FACS, fluorescence-activated cell sorting; FGCZ, Functional Genomics Center Zürich; GFP, green fluorescent protein; GO, Gene Ontology; IESC, intestinal epithelial stem cell; OCT, optimal cutting temperature; PC, principal component; Pdgfra, platelet-derived growth factor receptor A; rpm, revolutions per minute; RT-qPCR, quantitative reverse transcription PCR; SCENIC, single-cell regulatory network inference and clustering; scRNAseq, single-cell RNA sequencing; SMC, smooth muscle cell; TA, transit-amplifying cell; UMI, unique molecular identifier.

cells located at the bottom of the crypts in the small intestine and the colon [2]. Stem cell maintenance and proliferation requires multiple inputs from the surrounding microenvironment (the niche), such as Notch, epidermal growth factor (EGF), and Wnt signals [3]. These are provided by extra-epithelial cells or by the epithelium itself. We and several other groups showed that intestinal mesenchymal cells form the extra-epithelial, stem cell–supporting niche in the small intestine, and colon [4–9]. The subepithelial mesenchymal niche populations were defined by their ability to support the maintenance of intestinal epithelial stem cells (IESC) and secrete canonical Wnt ligands. However, the exact identity of these niche cells remained unclear since different, only partly overlapping, cell populations were described to act as the functional niche cells. They were differentially demarcated by platelet-derived growth factor receptor A (Pdgfra) [4], CD34 [5], Gli1 [6], Foxl1 [7], CD90 [8], and CD81 [9].

The advent of single-cell RNA sequencing (scRNAseq) has enabled researchers to investigate intestinal epithelial and mesenchymal cell heterogeneity with unprecedented resolution [10–12]. However, the applied approaches were based on specific cell isolation protocols that enriched solely for epithelial cells or the mesenchymal cells, leading to potential bias in the identifiable cell populations. Crypt-associated fibroblasts are tightly connected to the intestinal epithelium, thus mechanical or enzymatic removal of the epithelium may result in loss of certain mesenchymal subpopulations or vice versa.

In order to refine our understanding of the identity and function of the mesenchymal niche cells and their secreted factors, we performed an unbiased single-cell transcriptome analysis of the adult murine colon, looking at both epithelial and mesenchymal parts. We identify most of the known epithelial cell types and uncover the presence of two functionally distinct fibroblast populations: crypt-top fibroblasts (CTFs) and crypt-bottom fibroblasts (CBFs). These are also present in the colon of healthy humans. CBFs are $Pdgfra^{low}$ cells, located at the bottom of the crypts in close proximity to the stem cells and secrete canonical Wnt ligands (*Wnt2* and *Wnt2b*), Wnt-signaling-potentiators (*Rspo3*), and bone morphogenetic protein (Bmp) inhibitors (*Grem1*). CTFs are $Pdgfra^{high}$ cells, located at the top of the crypt and secrete noncanonical Wnt ligands (*Wnt5a*) and Bmp ligands (*Bmp2/3/4/5/7*), thereby inducing epithelial differentiation. Together, these two cell populations form the antagonistic gradients of canonical Wnt signaling and Bmp signaling required for the precise control of stem cell maintenance and epithelial differentiation.

# Results

## Unbiased single-cell profiling of murine colonic cells

To gain a deeper understanding of the heterogeneity and complexity of the colonic epithelium and mesenchyme, as well as the associated signaling networks, we performed scRNAseq of murine colonic tissue. We dissociated epithelial and mesenchymal fractions individually into single cells and used fluorescence-activated cell sorting (FACS) to further purify epithelial (EpCAM+, CD45−) and non-epithelial cells (EpCAM−, CD45−), while concurrently depleting lymphocytes (EpCAM−, CD45+) (Fig 1A and 1B). Overall, we observe a colonic cellular composition consisting of 91.2% epithelial cells and 8.8% non-epithelial cells. In contrast to previous studies, we next pooled the sorted epithelial (EpCAM+, CD45−) and non-epithelial (EpCAM−, CD45−) cells originating from both isolation protocols at equal fractions (1:1, epithelial:non-epithelial) to avoid epithelial overrepresentation and obtain sufficient data to describe both epithelial and non-epithelial cells in their full spectrum. Following quality control, scRNAseq results in a combined dataset of 7,395 cells with a mean transcript count of 3,243 per cell. Graph-based clustering reveals 16 individual clusters, which can be classified

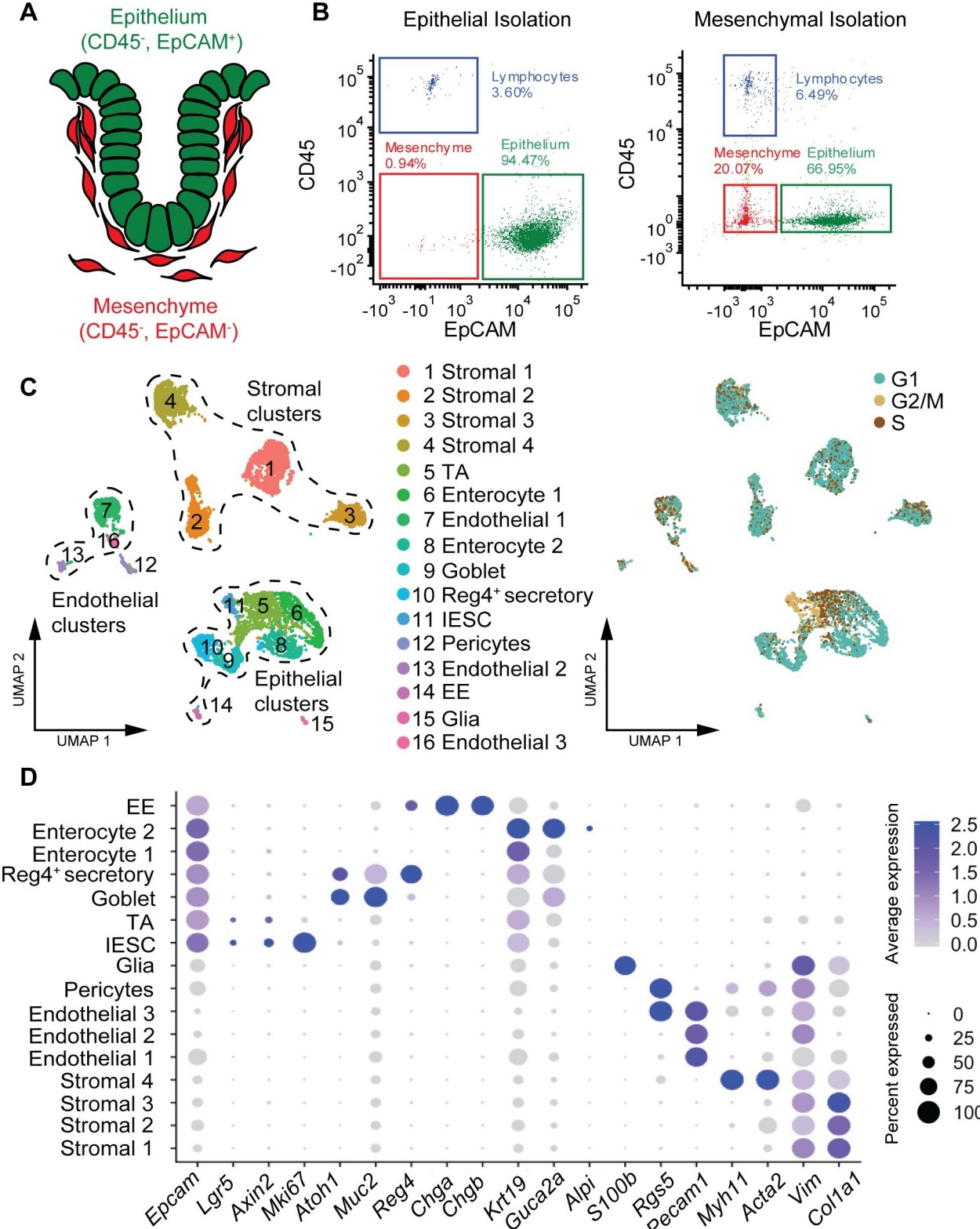

**Fig 1. Unbiased analysis of murine colon landscape reveals complexity and heterogeneity of epithelial and mesenchymal cells.** (A) Schematic representation of a colonic crypt with surface markers used to sort epithelial (green: EpCAM+, CD45−) and mesenchymal/non-epithelial cells (red: EpCAM−, CD45−). (B) Flow cytometry analysis using EpCAM and CD45 on single-cell suspension of colonic epithelial (left, raw data: S2 Data) and

mesenchymal cell isolation (right, raw data: S3 Data) prior to the sort. (C) Left: three major groups (epithelial, stromal, and endothelial) clustering into 16 distinct cluster revealed by UMAP analysis. Right: Epithelial cells show higher proliferative activity (cells in S-G2M) than non-epithelial cells (UMAP, single cells are colored according to their cluster annotation (left) or cell cycle phase (right)). (D) Characterization of cluster identity using relative expression of marker genes. (Dot plot, size, and color of the dot represent the percentage of cells which express the transcript and the average expression level within a cluster, respectively). EE, enteroendocrine; IESC, intestinal epithelial stem cell; TA, transit-amplifying cell; UMAP, Uniform Manifold Approximation and Projection.

into three major groups: Epithelium (*Epcam*⁺, *Krt19*⁺), stroma (*Vim*⁺, *Col1a1*⁺), and endothelium (*Pecam1*⁺) (Fig 1C, S1A Fig).

We could annotate most known epithelial cell types within our dataset with the exception of tuft cells. This designation is based on a combination of published marker genes [10,11,13] (Fig 1D) and differentially expressed genes (S1B Fig). Clusters 11 and 5 represent IESC and transit-amplifying cells (TA), respectively, based on the expression of *Lgr5*, *Axin2*, and *Mki67* (Fig 1D) in combination with their high proliferative activity (Fig 1C). Clusters 9 and 10 delineate the secretory lineage (*Atoh1*⁺), the former consisting of classic goblet cells (*Muc2*⁺) and the latter the recently described deep crypt secretory cells (*Reg4*⁺) [14]. Indeed, these cells appear to be mainly responsible for the activation of Notch signaling in intestinal stem cells as they express both ligands: *Dll1* and *Dll4* (S1D Fig) [15]. An absorptive lineage composed of clusters 6 and 8 (*Guca2a*⁺, *Alpi*⁺, *Aldh1l1*⁺) (Fig 1D, S1B Fig) consists of enterocytes strongly enriched for Gene Ontology (GO) terms such as "Generation of precursor metabolites and energy" and "Anion transport" (S1C Fig). The absorptive and secretory cells express a variety of Egf family ligands (*Egf*, *Tgfa*, *Areg*, *and Hbegf*) and ephrins (*Efnb1 and Efnb2*) (S1D Fig). Finally, cluster 14 comprises enteroendocrine cells (*Chga*⁺, *Chgb*⁺, and *Neurod1*⁺) which express high levels of *Ihh*—the main hedgehog ligand secreted by the intestinal epithelium (S1D Fig) [16].

We observed four stromal clusters (clusters 1, 2, 3 and 4), which all express typical fibroblast markers, such as *Vim* and *Col1a1*. In addition, we found clusters 12 and 15, which are pericytes (*Rgs5*⁺) and glia (*S100b*⁺), respectively.

Taken together, these data represent a comprehensive, high-quality dataset of murine colonic cell diversity, comprised of matching epithelium and mesenchyme.

## Characterization of crypt-associated fibroblasts

We next subset and reanalyzed the four stromal clusters (stromal 1 to 4, Fig 1C) to gain insight into distinct subpopulations of crypt-associated fibroblasts and their individual niche potential in more detail (Fig 2A). Cluster S4 cells are likely smooth muscle cells (SMCs) due to the expression of *Acta2* and *Myh11* (see Fig 2D) in combination with GO term enrichment for muscle cell-specific functions, e.g., "Regulation of skeletal muscle contraction by calcium ion signaling" and "Myofibril assembly" (S2A Fig). However, we cannot exclude that this cluster also contains myofibroblasts.

We interrogated the expression of markers previously used to demarcate the cells of the murine stem cell niche (Fig 2B). The hedgehog transcription factor *Gli1* [6] shows ubiquitous expression in all stromal clusters (S1 to S4). *Cd34* [5] and *Cd90* [8] display a strong enrichment in clusters S1 and S2. However, *Cd90* is also expressed in SMC and *Cd34* is additionally expressed in endothelial cells. *Foxl1* [7,17] is expressed specifically in clusters S3 and S4. Interestingly, while *Pdgfra* [4] is expressed in clusters S1, S2, and S3, it displays a strong difference in transcript levels. Indeed, cluster S3 shows a high expression of *Pdgfra* and clusters S1 and S2 show a low expression of *Pdgfra* (Fig 2C). Due to differences in spatial localization of Pdgfra-positive cells in regards to the colonic crypt (see below, Fig 3B and 3C), we henceforth refer to cluster S3 as CTFs and to clusters S1 and S2 as CBFs (CBFs1 and CBFs2, respectively). In

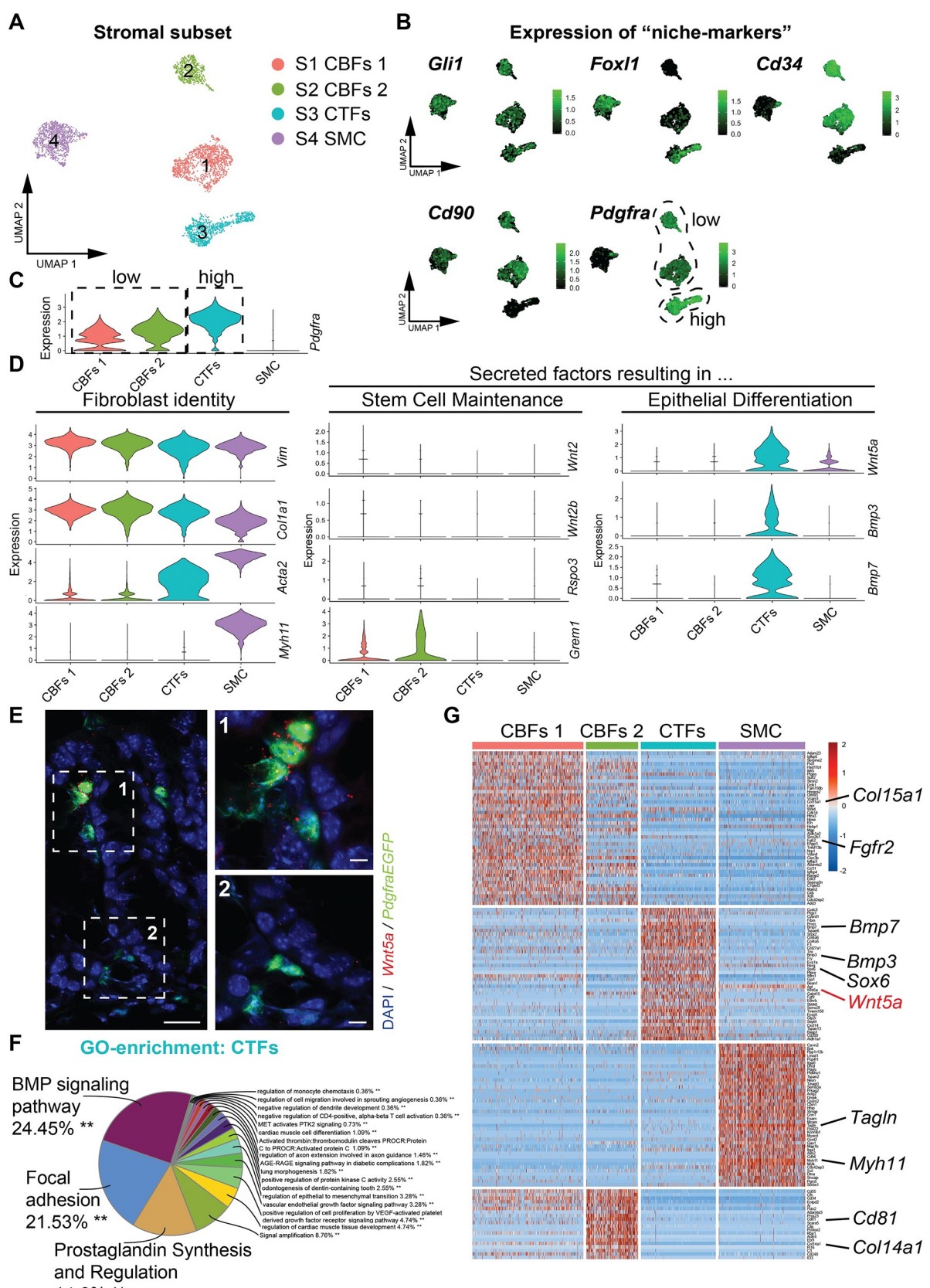

**Fig 2. CTFs (*Pdgfra^high^*) and CBFs (*Pdgfra^low^*) mark distinct functional colonic mesenchymal cell populations that control stem cell proliferation and epithelial differentiation in the murine colon.** (A) Heterogeneity of colonic mesenchymal cells (UMAP plot of the reanalyzed stromal subset, single cells are colored according to cluster annotation). (B) Relative expression of niche marker genes (UMAP, single cells are colored according to transcript expression). (C) Pdgfra expression across stromal clusters (violin plot, raw data: S1A Data). (D) Relative expression of fibroblast identity marker genes (*Vim*, *Col1a1*, *Acta2*, and *Myh11*), secreted factors resulting in stem cell maintenance (*Wnt2*, *Wnt2b*, *Rspo3*, and *Grem1*) and secreted factors resulting in epithelial differentiation (*Wnt5a*, *Bmp3*, and *Bmp7*) (violin plots, raw data: S1B Data). (E) *Wnt5a* is expressed by CTFs at the top of the colonic crypts (single-molecule RNA in situ hybridization (Scale bar = 20 μm, 5μm (1,2)). (F) GO enrichment terms for CTFs. (G) Heatmap of genes that are differentially expressed among the mesenchymal clusters (0.95 quantile) show strong similarities between CBFs1 and CBFs2 and clear distinction of CTFs and SMC. Bmp, bone morphogenetic protein; CBF, crypt-bottom fibroblast; CTF, crypt-top fibroblasts; GO, Gene Ontology; Pdgfra, platelet-derived growth factor receptor A; SMC, smooth muscle cell; UMAP, Uniform Manifold Approximation and Projection.

accordance with their localization at the bottom of the crypt in close proximity to the intestinal stem cells, CBFs express factors important for stem cell maintenance, such as canonical Wnt ligands (*Wnt2* and *Wnt2b*), Wnt signaling potentiators (*Rspo3*), and Bmp signaling antagonists (*Grem1*) (Fig 2D, S2B Fig). In contrast, CTFs express factors resulting in the induction of epithelial differentiation, such as Bmp ligands (*Bmp2*, *Bmp3*, *Bmp4*, *Bmp5*, and *Bmp7*), noncanonical Wnt ligands (*Wnt5a*), and inhibitors of canonical Wnt signaling (*Dkk3*) (Fig 2D and 2E, S2B Fig).

GO term enrichment recapitulates the molecular difference between crypt-associated fibroblast populations, with CTFs showing an enrichment for "Bmp signaling pathway", "Focal adhesion," and "Prostaglandin Synthesis and Regulation" (Fig 2F), while CBFs show enrichment for terms such as "response to vascular endothelial growth factor signaling" and "regulation of extracellular matrix" (S2C and S2D Fig).

In sum, the analysis of differentially expressed genes exposes a high similarity in marker expression between the clusters CBFs1 and CBFs2 and a clear demarcation of CTFs and SMC (Fig 2G). However, we find interesting smaller differences between CBFs1 and CBFs2. CBFs1 (Fig 2G) show higher, but not exclusive, expression of the canonical Wnt ligands *Wnt2* and *Wnt2b* (S3B Fig). In addition, they express *Igfbp3*, *Igfbp5*, *Adam23*, and *Edil1* (Fig 2G), pointing toward potent integrin/extracellular signal-regulated kinase (ERK) signaling regulation [18]. CBFs2 (Fig 2G) show higher, but not exclusive, expression of the Bmp antagonist *Gremlin 1* and the marker *Cd81* (Fig 2G, S3B Fig), displaying high similarity to the recently described small intestinal trophocytes [9].

Furthermore, we employed the CellphoneDB [19] algorithm to infer possible communication (ligand–receptor interactions) between CBFs1, CBFs2, CTFs, and epithelial cells (S3A Fig). We were able to confirm expected crosstalk between populations, such as Wnt5a (from CTFs) potentially signaling to the noncanonical Wnt receptors Ror1 (expressed in cells of cluster Enterocyte 1) and Ror2 (expressed in IESC, Goblet, Reg4+ secretory, and enteroendocrine cells). Similarly, Wnt2 (from CBFs1, CBFs2) would signal to canonical Wnt receptor Fzd3 (expressed in TA, Goblet, and enteroendocrine) (S3A Fig). In addition, we uncovered some interesting new potential crosstalk, such as between Fgf2 (from CBFs1 and CBFs2) and Cd44 (expressed in IESC and TA cells) and Fgfr2 (expressed in IESC, TA, Goblet, Reg4+ secretory, and enteroendocrine cells) (S3A Fig). Intriguingly, CBFs2 express the ligands Igf1, Efna2, Efna5, and Ptn whose receptors Igf1r (Igf1), Epha4 (Efna2, Efna5), and Ptprs (Ptn) are expressed on epithelial cells (S3A Fig). The relevance of this crosstalk needs to be explored.

Intriguingly, different Wnt signaling antagonists (*Sfrp1*, *Sfrp2*, *Frzb*, *Sfrp4*, and *Dkk3*) are expressed in distinct crypt-associated fibroblasts. This may serve to sharpen the gradient of canonical Wnt signaling activity (S2B Fig).

We also performed gene regulatory network analysis using single-cell regulatory network inference and clustering (SCENIC) [20]. This revealed cluster specific enrichment for transcription factor regulons. CTFs show an enrichment for the regulon activities of Lef1, Runx1,

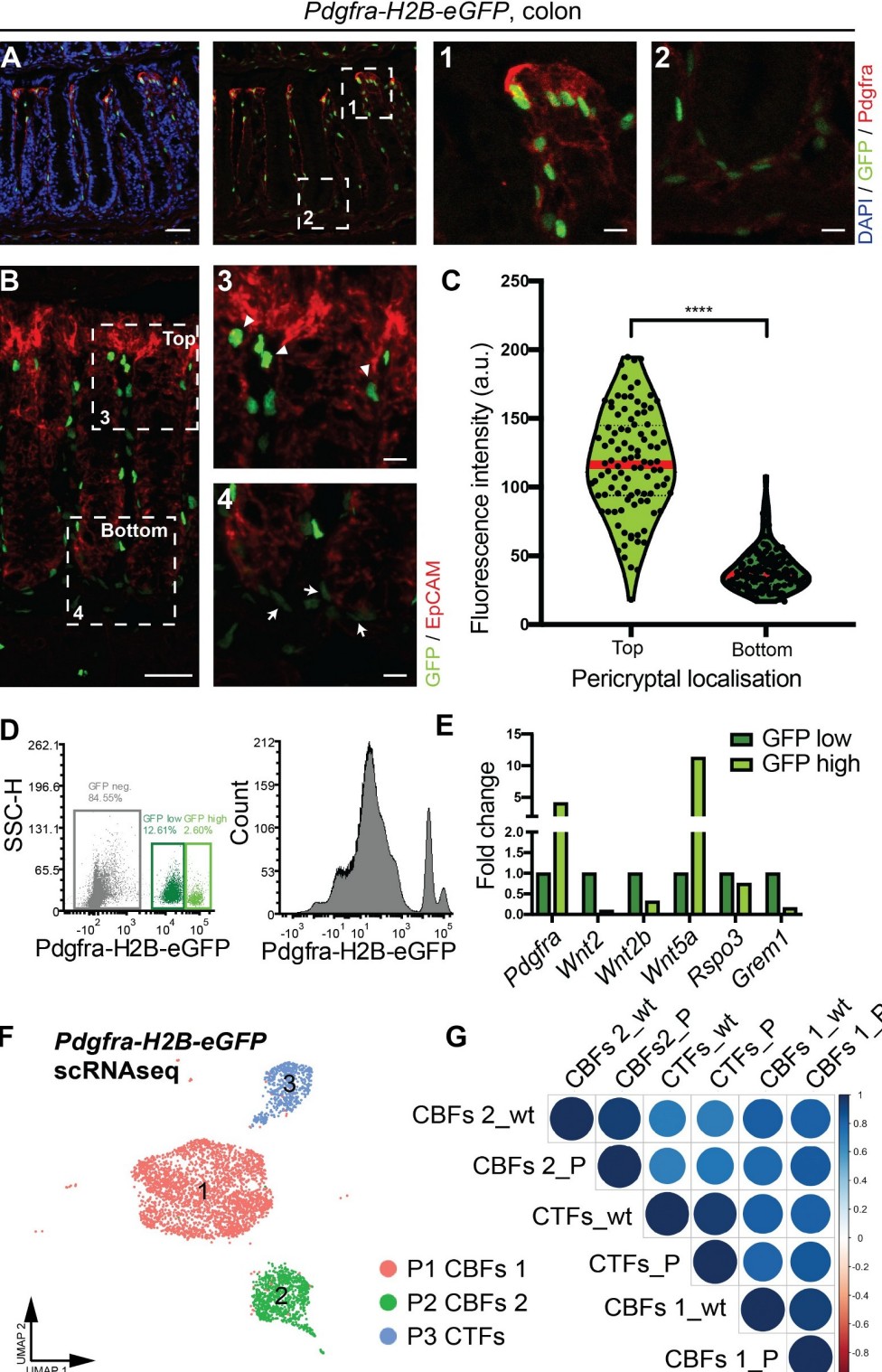

**Fig 3. CTFs ($Pdgfra^{high}$) and CBFs ($Pdgfra^{low}$) are localized in a distinct manner along the colonic crypt axis.** (A–C) Cryosections of *Pdgfra-H2B-eGFP mice*. (A) High Pdgfra protein expression correlates with the level of *Pdgfra* transcript in *Pdgfra-H2B-eGFP* reporter mouse line. (Red: Pdgfra protein, green: GFP, blue: DAPI) (Scale bar = 40 μm), (1,2) Insets of crypt top and crypt bottom respectively (Scale bar = 5 μm). (B) CTFs ($Pdgfra^{high}$) are localized at the top of the colonic crypt, whereas CBFs ($Pdgfra^{low}$) are at the bottom. (Red: EpCAM, green: GFP, blue:

DAPI) (Scale bar = 40 μm). (3,4) Insets of crypt top and crypt bottom, respectively, arrowheads point out *Pdgfra^high^* (GFP high) cells and arrows point out *Pdgfra^low^* (GFP low) cells. (Scale bar = 5 μm) (C) Fluorescence intensity measurement of GFP positive nuclei. $n_{Bottom}$ = 159, $n_{Top}$ = 100, **** = $p < 0.0001$ (Unpaired *t* test with Welch correction) (Raw data: S1C Data) (D) Flow cytometry analysis for GFP on mesenchymal cells isolated from *Pdgfra-H2B-eGFP* mice identifying GFP−, GFP-low, and GFP-high cells (Raw data: S4 Data). (E) Relative mRNA expression of *Pdgfra*, *Wnt2*, *Wnt2b*, *Wnt5a*, *Rspo3*, and *Grem1* in CTFs (GFP high) and CBFs (GFP low) isolated from *Pdgfra-H2B-eGFP* reporter mouse line. (RT-qPCR, normalized to *Hprt* expression levels, FC levels in CBFs set to 1) (Raw data: S1D Data). (F) CBFs1, CBFs2, and CTFs populations determined by scRNAseq from colonic *Pdgfra*-positive cells (*Pdgfra-H2B-eGFP* reporter mouse line). (UMAP, single cells are colored according to cluster annotation). (G) Stromal clusters from wt and *Pdgfra*+ colonic scRNAseq analysis display high similarity. (Spearman correlation). CBF, crypt-bottom fibroblast; CTF, crypt-top fibroblasts; FC, fold change; GFP, green fluorescent protein; Pdgfra, platelet-derived growth factor receptor A; RT-qPCR, quantitative reverse transcription PCR; scRNAseq, single-cell RNA sequencing; UMAP, Uniform Manifold Approximation and Projection; wt, wild-type.

Nkx2-3, and Foxl1 (S2E Fig). The transcription factor Nkx2-3 had previously been found to control *Bmp2/4* expression in the small intestine [21]. Lef1 regulon activity in CTFs may hint at a possible explanation for the presence of the recently discovered small intestinal *Lgr5*+ villus tip telocytes [22], given that *Lgr5* is canonical Wnt targe gene [23], and Lef1 is both a canonical Wnt target gene and a Wnt/b-catenin transcription factor [24]. Intriguingly, this would also point to a difference in transcriptional regulation of *Lgr5*+ mesenchymal cells compared to *Lgr5*+ intestinal stem cells, which have been shown to mainly depend on TCF family member Tcf4 [25].

In conclusion, we identified two molecularly distinct colonic crypt-associated fibroblast populations, which are defined by their potential to either support the maintenance of epithelial stem cells or to induce the differentiation of the intestinal epithelium.

## Spatial localization of crypt-associated fibroblasts

Finally, we sought to identify the position of the molecularly defined crypt-associated fibroblast populations along the colonic crypt axis. We analyzed colonic tissue sections of *Pdgfra-H2B-eGFP* reporter mice [26]. These display differences in nuclear green fluorescent protein (GFP) intensity closely correlated with the *Pdgfra* transcript levels and perfectly recapitulate differences in Pdgfra protein expression in the murine colon, as exemplified by the co-localization of nuclei with high GFP levels and cells exhibiting high intensity Pdgfra antibody staining (Fig 3A). All cells with GFP positive nuclei show subepithelial localization and co-expression with the pan-fibroblast maker Vimentin (S4A Fig). Fluorescence intensity quantification of GFP positive nuclei revealed a clear association between the localization of fibroblasts and nuclear GFP intensity. CTFs displayed high nuclear GFP signals (mean + SD: 117.1 + 37.9), and CBFs displayed low nuclear GFP signals (mean + SD: 39.5 +14.7) (Fig 3B and 3C). To further investigate the morphology of Pdgfra-expressing cells in the colon, we examined colon from mice carrying the *Pdgfra-Cre^ERT2^* driver [27] in combination with the cytoplasmic fluorescent reporter mouse line *Ai14* (*LSL-tdTomato*) [28] (single tamoxifen injection, analysis 1 day post injection). As has been previously observed in telocytes of the small intestine [22], we find that *Pdgfra*-expressing cells have elongated processes and small telopode portrusions (S4B Fig). The morphology is revealed by the distribution of tdTomato.

To validate the gene expression patterns found in our scRNAseq (Fig 2), we isolated colonic mesenchymal GFP-high and GFP-low cells from the *Pdgfra-H2B-eGFP* mouse line (Fig 3D, S4C Fig). Quantitative real-time PCR confirmed higher expression of *Wnt2*, *Wnt2b*, *Rspo3*, and *Grem1* in GFP-low cells compared to GFP-high cells, recapitulating the differences observed between CBFs and CTFs in our scRNAseq (Fig 3E). As expected, *Wnt5a* and *Pdgfra* expression was higher in the GFP-high cells (Fig 3E).

In order to further corroborate our findings, we performed an independent scRNAseq analysis of GFP positive colonic mesenchymal cells of the *Pdgfra-H2B-eGFP* mouse line. Graph-based clustering revealed three main clusters (Fig 3F). Based on the expression of *Pdgfra* and various niche factors (*Wnt2/2b*, *Grem1*, *Rspo3*, and *Wnt5a*), we designated cluster P3 as CTFs and cluster P1 and P2 as CBFs1 and CBFs2 (S4D and S4E Fig). SMCs (possibly including myofibroblasts) were not found in this dataset. Based on the top 2,000 differentially expressed genes in each cluster, CBFs1, CBFs2, and CTFs originating from the *Pdgfra-H2B-eGFP* scRNAseq analysis show very high similarity (Spearman correlation) to the three Pdgfra positive clusters of the same name in our initial scRNAseq experiment (Fig 3G).

Taken together, these data confirm that CTFs and CBFs correspond to Pdgfra high and low expressing cells, respectively and are differentially localized along the crypt axis.

## Murine colonic crypt-associated fibroblast identities are conserved in human healthy colon

In order to see whether the molecular identity of murine CBFs and CTFs are evolutionarily conserved, we compared our findings to human colonic stromal cell populations. To do so, we took advantage of the recently published dataset (GSE114374), in which human colonic stroma has been analyzed in inflammatory disease progression versus healthy state at single-cell level [12]. We performed a bioinformatic reanalysis of Kinchen's healthy colonic stromal dataset. We find a total of 8 clusters, which we annotated, based on published markers [12] as fibroblasts, plasma cells, glia, endothelial cells, SMCs, and pericytes (Fig 4A). We interrogated the expression of previously published murine niche marker genes within the fibroblast clusters (Fig 4B). A very similar picture emerges to that seen in mouse (Fig 2B), with the exception of *CD34*, which is expressed solely in endothelial cells in the human colon.

Interestingly, reinforcing our observations in the murine colon, the strength of *PDGFR*A expression can be used to demarcate distinct crypt-associated fibroblast populations in the human colon as well (Fig 4C). Taken together with the expression pattern of components of the WNT and BMP signaling pathways (Fig 4D), we identify clusters HC1 and HC3 as *PDGFRA*$^{low}$ CBFs1 and CBFs2, respectively, that express canonical WNT ligands (*WNT2B*), WNT signaling potentiators (*RSPO3*), and BMP inhibitors (*GREM1*, *GREM2*, and *CHRD*). In addition, using the R package matchSCore2 [29], we compared the murine mesenchymal populations with the human mesenchymal populations on the level of the entire transcriptome. We found high interspecies overlap between murine and human CBFs, murine and human CTFs, and murine and human SMCs (S5 Fig).

However, diverging from the expression patterns observed in the murine colon, the canonical WNT ligand *WNT2B* is expressed not only in CBFs, but also additionally in SMCs (Fig 4D). This finding is consistent with a recent observation that embryonic SMCs express *WNT2B* during human intestinal development [30]. Moreover, we identify cluster HC2 as *PDGFRA*$^{high}$ CTFs, expressing a broad spectrum of BMP ligands (e.g., *BMP2*), and noncanonical WNT ligands (e.g., *WNT5A*) (Fig 4D). Interestingly, just like in the murine colon, we also find expression of various canonical WNT signaling inhibitors (*SFRP1*, *SFRP2*, *FRZB*, and *DKK3*) in all the fibroblast clusters in the human colon; however, compared to mouse, the positions of *DKK3* and *FRZB* expression seems to be reversed in human CBFs and CTFs (Fig 4D and S2B Fig).

Future studies of this highly specific spatial localization and molecular regulation might reveal deeper mechanistic insights into how canonical WNT ligands and antagonists interact to create a canonical WNT signaling gradient optimal for intestinal stem cell maintenance and proliferation.

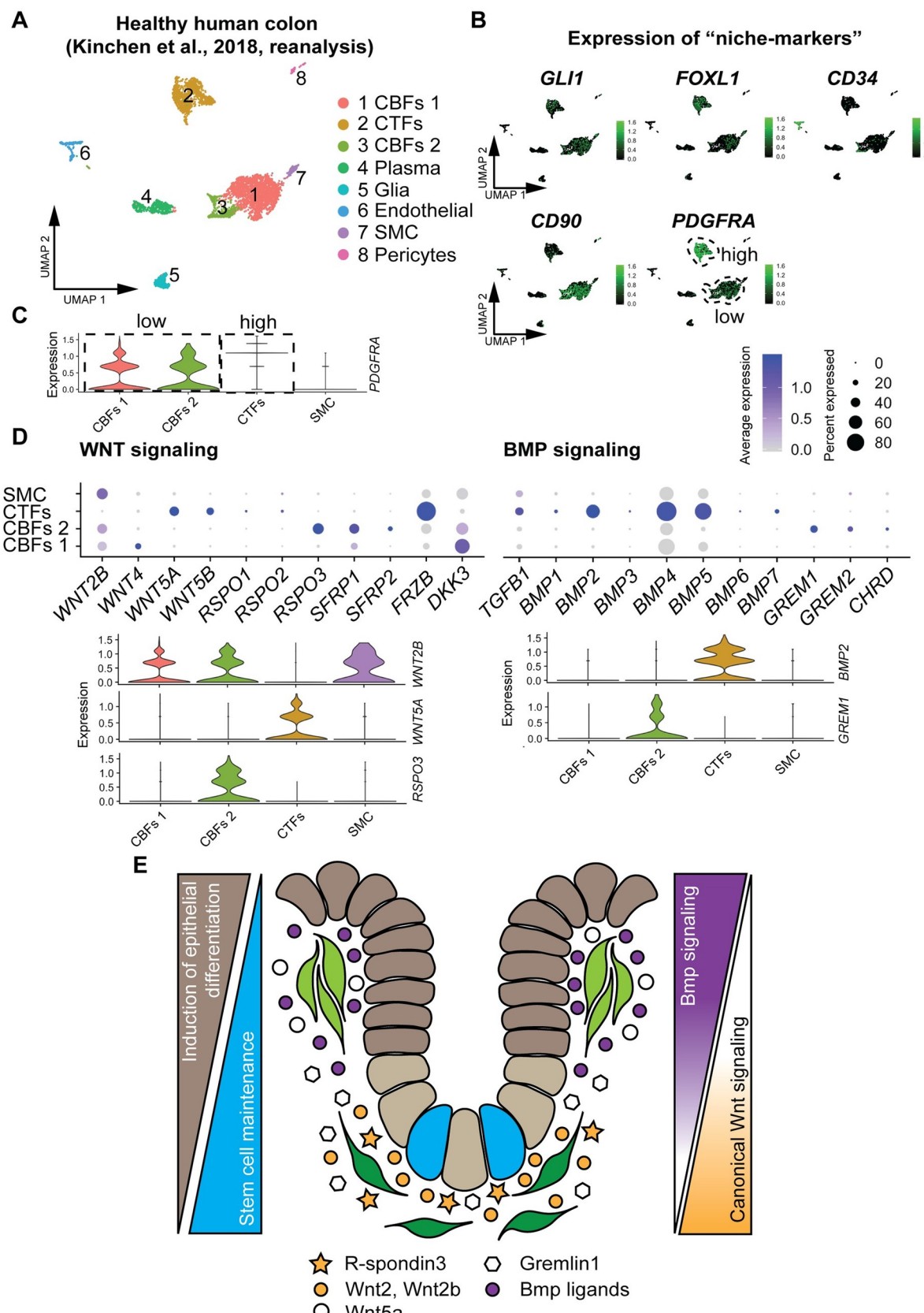

**Fig 4. Murine CTFs and CBFs populations are conserved in healthy human colon.** (A) Reanalysis of the healthy colonic stromal dataset from GSE114374 reveals landscape similar to murine colon. (UMAP, single cells are colored according to cluster annotation). (B) Conserved niche markers determine analogous cell populations in human colonic mesenchyme (UMAP, single cells are colored according to transcript expression). (C) Relative expression of *PDGFRA* separates human mesenchymal cells into three *PDGFRA*[+] populations: CTFs, CBFs 1, and CBFs 2. (Violin plot, raw data: S1E Data) (D) Relative expression of factors involved in WNT and BMP pathways. (Violin and dot plots, size, and color of the dot represent the percentage of cells which express the transcript and the average expression level within a cluster, respectively; raw data: S1F Data). (E) Schematic model of conserved colonic crypt-associated fibroblasts. *PDGFRA*[high] CTFs secrete BMP ligands and noncanonical WNT ligands, resulting in epithelial differentiation. *PDGFRA*[low] CBFs secrete canonical WNT ligands, WNT potentiators, and BMP inhibitors to maintain stem cells (blue). BMP, bone morphogenetic protein; CBF, crypt-bottom fibroblast; CTF, crypt-top fibroblast; PDGFRA, platelet-derived growth factor receptor A; UMAP, Uniform Manifold Approximation and Projection.

## Discussion

Understanding the cellular and molecular architecture of the colon is crucial to comprehend how it functions in homeostasis and to find suitable strategies to treat colonic diseases. Colorectal cancer and inflammatory bowel disease are important disorders that pose a great challenge to modern medicine due to their widespread and complex pathology. In recent years, it has been shown that crosstalk between fibroblasts and epithelial cells plays an important role in the initiation and progression of these diseases [31,32].

In this study, we provide a comprehensive picture of the murine colonic cellular diversity and investigate in detail the identity of mesenchymal niche cells. We produced a high-quality single-cell transcriptome atlas of matched colonic epithelium and mesenchyme, creating a powerful resource to study cellular identities and pathway regulation during adult tissue homeostasis. We propose a novel classification of murine and human crypt-associated fibroblasts, based on their position relative to the crypt axis and their molecular identity (Model Fig 4E). *Pdgfra*[high] CTFs are located at the top of the crypt and secrete Bmp ligands (*Bmp2/3/4/5/7*) and noncanonical Wnt ligands (*Wnt5a*), thereby inducing differentiation in the nearby epithelial cells. *Pdgfra*[low] CBFs are located in close proximity to the intestinal stem cells at the bottom of the crypt. They secrete factors integral to stem cell maintenance, such as canonical Wnt ligands (*Wnt2* and *Wnt2b*), Wnt signaling potentiators (*Rspo3*), and Bmp inhibitors (*Grem1*). Wnt2b has previously been shown to support the growth of intestinal organoids [33]. Expression of Pdgfrα has recently been described to be a useful marker for pericryptal stromal cells that act as source for Wnts and Rspo3 in the murine intestinal system [4]. Our work shows that this marker demarcates a second, additional signaling hub at the crypt top.

CTFs share characteristics with previously described intestinal telocytes, such as expression of *Sox6* and *F3* [12], as well as high expression of *Foxl1* likely responsible for the high Foxl1 regulon activity described above [7,17]. The high expression of *Wnt5a* in CTFs is intriguing. Taken together with the expression of Bmp ligands (e.g., *Bmp2* and *Bmp4*) and marker gene *F3*, the data suggest that the colonic CTFs are likely analogous to the villus tip telocytes observed in the small intestine [22]. Wnt5a has been show to inhibit canonical Wnt signaling by down-regulating transcription of Wnt target genes [34,35], which is consistent with the proposed role of CTFs in orchestrating epithelial differentiation. Upon intestinal injury, Wnt5a is required for the proper subdivision of wound channels and thereby the formation of new crypts [36]. During embryogenesis, Wnt5a is involved in the clustering of the mesenchymal core below prospective epithelial invaginations—an integral process for crypt formation in the developing gastrointestinal system [37]. The role of Wnt5a in homeostasis of this tissue remains to be explored.

CBFs co-express most previously discussed stem cell niche-cell markers, such as *Gli1* [6], *Pdgfra* [4], *Cd90* [8], and *Cd34* [5]. Importantly though, none of these markers is truly definitive. *Gli1* and *Pdgfra* are expressed in all intestinal crypt-associated fibroblasts; *Cd90* and *Cd34* are additionally expressed in SMCs and endothelial cells, respectively. Ultimately, it is the combination of these markers that delineate CBFs and thereby the stem cell niche.

Together, CBFs and CTFs generate the antagonistic gradients of canonical Wnt signaling and Bmp signaling, which dictate the zones of epithelial proliferation and differentiation [3]. In the small intestine, the Bmp signaling gradient has recently been described to be formed via ligand secretion by villus telocytes and antagonist secretion by *Cd81*+ trophocytes [9]. Our observations about CTFs and CBFs allow to draw a similar picture in the colon, where the two fibroblast populations, which are distinguishable by their level of Pdgfra expression, produce either Bmp signals (CTFs) or Bmp inhibitors (CBFs). There are striking differences between the small intestine and the colon, both in morphology and niche signals. The epithelium of the small intestine consists of crypts and villi, whereas only crypts are found in the colonic epithelium. Canonical Wnt ligands, which are integral to intestinal epithelial stem cell maintenance, are redundantly secreted by epithelial Paneth cells [38], as well as mesenchymal cells [4–7,17] in the small intestine. In the colon, there is no epithelial Wnt source. Accordingly, the requirement for the mesenchymal niche (e.g., canonical Wnt signals) is not necessarily the same in colon and small intestine. However, combined, the data produced by us and McCarthy and colleagues [9] show a strong conservation in intestinal fibroblast identity and corroborates the validity of using Pdgfra expression levels as a marker to distinguish stem cell–supporting fibroblasts from fibroblasts that induce epithelial differentiation.

The difference in Pdgfra expression level as a marker for fibroblast populations has been described in the lung, where it serves to discriminate between lipofibroblasts and myofibroblasts [39–41]. We note a correlation between high *Pdgfra* expression and *Acta2* expression in CTFs (Fig 2D), pointing toward a possible myofibroblast identity of these cells. However, in lung fibroblasts, there are conflicting findings—both correlation [39] and anticorrelation [40]—have been reported. Ultimately, it remains to be elucidated, both in the lung and the gastrointestinal system, whether the different *Pdgfra* expression levels control the identity and ultimately function of the two fibroblast populations.

In conclusion, we propose here a novel classification of intestinal crypt-associated fibroblasts: CBFs and CTFs. It will be imperative to investigate the roles of these two fibroblast populations in the context of tumor initiation, progression, and metastasis.

## Materials and methods

### Ethics statement

All mouse procedures were performed in accordance with Swiss Guidelines and approved by the Cantonal Veterinary Office Zürich, Switzerland (ZH156/2015 and ZH169/2016).

### Mice

C57BL/6 wild-type mice were purchased from Charles River, Germany (Strain code: 027). *Pdgfra*^H2BeGFP [26], *Pdgfra-Cre*^ERT2 [27], and *Ai14* [28] mice were purchased from Jackson Laboratories, United States of America (Stock numbers: 007669, 032770, and 007914). Cells were isolated from female mice at age 8 to 12 weeks.

### Epithelial and mesenchymal cell isolation and cell sorting

Colonic epithelial and mesenchymal cells were isolated inspired by previous publications [42,43] and modified as described below. Colonic tissue was harvested, minced into small pieces (2 mm), and subsequently washed with PBS. Tissue pieces were then incubated in Gentle Cell Dissociation Reagent (STEMCELL Technologies, Germany) for 30 minutes at room temperature to detach the epithelium. The epithelial fraction was then filtered through a

Falcon 70-μm cell strainer (Corning, Switzerland), washed with plain ADMEM/F12 and incubated for 5 minutes at 37˚C in prewarmed TrypLE express (Gibco, Thermofisher, Switzerland), followed by single-cell dissociation using the m_intestine program on the gentleMACS Octo Dissociator (Miltenyi Biotec, Switzerland). The epithelial single-cell suspension was then filtered through a Falcon 40-μm cell strainer (Corning) and stored on ice in ADMEM/F12 (supplemented with 10% FBS). After the detachment of the epithelium, the remaining tissue pieces were digested for 1 hour at 37˚C under 110 revolutions per minute (rpm) shaking conditions in DMEM supplemented with 2 mg/mL collagenase D (Roche) and 0.4 mg/mL Dispase (Gibco). The mesenchymal fraction was then filtered through a Falcon 70-μm cell strainer (Corning), washed with plain ADMEM/F12, and subsequently filtered through a Falcon 40-μm cell strainer (Corning). Both epithelial and mesenchymal cells were stained for 30 minutes on ice with anti-CD45-PE (1:500, eBioscience/Thermofisher, Switzerland) and anti-CD326 (EpCAM)-FITC (1:500, eBioscience/Thermofisher, Switzerland) in ADMEM/F12 (supplemented with 10% FBS). Prior to cell sorting, all cells were stained for 5 minutes on ice with Zombie Violet Fixable Viability Kit in PBS (1:1000, Biolegend, Thermofisher, Switzerland). In the case of mesenchymal cell isolation from $Pdgfra^{H2BeGFP}$, mice only staining with Zombie Violet Fixable Viability Kit was performed. Cells were detected and sorted at the Cytometry Facility at the University of Zürich using a FACSAria III cell sorter (gates visible in corresponding figures) (BD Biosciences, Switzerland).

## Single-cell RNA sequencing

Colonic epithelial and mesenchymal cells (Fig 1) were isolated from three female C57BL/6 mice as described above and mixed at a ratio of 1:1. Using the Single Cell 3' V3 assay, 16,000 cells were loaded on a Chromium Controller (10x Genomics, USA). Colonic mesenchymal Pdgfra-positive cells (Fig 3) were isolated as described above from three female $Pdgfra^{H2BeGFP}$ mice. Using the Single Cell 3' V2 assay, 16,000 cells were loaded on a Chromium Controller (10x Genomics). In both cases reverse transcription, cDNA synthesis/amplification and library preparation were carried out in accordance with the manufacturer's recommendations by the Functional Genomics Center Zürich (FGCZ). scRNA libraries were sequenced on a NovaSeq 6000 instrument (Illumina, Switzerland).

## Tissue isolation, cryosectioning, and immunohistochemistry

Colonic tissue was harvested from female $Pdgfra^{H2BeGFP}$ mice, fixed for 1 hour at room temperature in 4% paraformaldehyde in PBS (ChemCruz/Santa Cruz Biotechnology, USA), and incubated O/N in 30% sucrose in PBS at 4˚C. Tissue was then kept in a 1:1 mix of 30% sucrose and optimal cutting temperature (OCT) (TissueTek/Biosystems Switzerland) for 30 minutes at 4˚C, before embedding in OCT and being cooled to −80˚C. OCT-embedded tissue was cryosectioned using a Microm HM560 Cryostat (Thermo Fisher Scientific, Switzerland) at 5 μm, dried for 2 hours at room temperature before either being directly used for immunohistochemistry, or stored at −80˚C. Standard immunohistochemical protocols were performed with the following primary antibodies (1:100 dilution): mouse-anti-Acta2 (Sigma, Germany), goat-anti-Pdgfra (R&D Systems, USA), chicken-anti-Vimentin (Millipore, USA), and rabbit-anti-EpCAM (Abcam, UK). Secondary antibodies (1:400 dilution) were anti-rabbit, anti-mouse, anti-goat antibodies conjugated with Alexa Dyes (A555, A598, or A647) (Thermo Fisher Scientific, Switzerland). Sections were counterstained with DAPI, mounted with Fluor-Save reagent (Sigma), imaged on a Leica SP8 laser scanning confocal microscope (Leica, Switzerland), and subsequently processed using ImageJ (FIJI).

## In situ RNA hybridization

mRNAs were localized in specific cells using the RNAscope method (Advanced Cell Diagnostics, Germany) on colonic tissue sections according to manufacturer's instructions (RNAscope Fluorescent Multiplex Assay). Probe sets for *Wnt5a* was designed by Advanced Cell Diagnostics. Images were obtained on a Leica SP8 laser scanning confocal microscope.

## RNA isolation, cDNA synthesis, and quantitative real-time PCR

Total RNA from Pdgfra-high and Pdgfra-low cells (Fig 4) was isolated using the mirVana miRNA Isolation Kit (Thermo Fisher Scientific) according to the manufacturer's instructions. cDNA was synthesized using the RNA to cDNA EcoDry synthesis kit (TaKaRa/Thermo Fisher Scientific, Switzerland). Quantitative real-time PCR reactions were performed with technical triplicates using the SYBR Green Kit (Applied Biosystems/Thermo Fisher Scientific, Switzerland) and monitored by the QuantStudio3 system (Applied Biosystems). Sequences of the used primers are: *Hprt*: AAG CTT GCT GGT GAA AAG GA, TTG CGC TCA TCT TAG GCT TT; *Pdgfra*: TCC TTC TAC CAC CTC AGC GAG, CCG GAT GGT CAC TCT TTA GGA AG; *Wnt2*: CTG AGT GGA CTG CAG AGT GC, ACA ACG CCA GCT GAA GAG AT; *Wnt2b*: CAC CCG GAC TGA TCT TGT CT, TGT TTC TGC ACT CCT TGC AC; *Wnt5a*: CAG GGT GAT GCA AAT AGG CAG, AGC CAT AGT CGA TGT TGT CTC C; *Rspo3*: ACT ACA GCA TCC TTC AGC C, TTT TCG TTT TCT CTC TCT TCC C; and *Grem1*: ACC CAG AGT ACC GTG GT, GTG TAT GCG GTG CGA TTC A.

## Computational analysis for scRNA-seq

**Preprocessing.** The sequencing libraries were de-multiplexed, aligned to the mouse transcriptome (mm10), and unique molecular identifiers (UMI) were counted using Cell Ranger (10x Genomics) version 3.0.1 by the FGCZ. Further data analysis was performed using the Seurat package version 3.1 [44] in R version 3.6.2. For the in-depth analysis of colonic cell populations (Fig 1C), cells with counts in more than 200 genes and genes that were detected in more than 5 cells were retained, resulting in 10,427 cells and 17,956 genes. Following a second round of quality control, only cells with counts in 500 to 7,500 genes and <25% mitochondrial genes were retained, resulting in a final dataset consisting of 7,395 cells. Pdgfra⁺ cells (Fig 3F) with counts in more than 200 genes and genes that were detected in more than 5 cells were retained, resulting in 4,500 cells and 16,055 genes. Following a second round of quality control, only cells with counts in 500 to 4,000 genes and <10% mitochondrial genes were retained, resulting in a final dataset consisting of 4,409 cells. For the reanalysis of the human colonic mesenchymal cell dataset [12] (Fig 4A), cells with counts in more than 200 genes and genes that were detected in more than 5 cells were retained, resulting in 4,369 cells and 17,652 genes. Following a second round of quality control, only cells with counts in 500 to 4,000 genes were retained, resulting in a final dataset consisting of 4,302 cells.

**Dimensionality reduction and clustering.** For all three datasets, normalization, scaling, and variable gene selection was performed using the "SCTransform" function in Seurat [45] with standard settings. Following principle component analysis, dimensionality reduction using Uniform Manifold Approximation and Projection (UMAP) algorithm [46] was performed using 30 principal components (PCs). Clusters were then identified using the "FindNeighbors" and "FindClusters" functions in Seurat with resolution parameters of 0.45 (Fig 1C) and 0.3 (Figs 3F and 4A). Clusters were then annotated based on known epithelial and stromal marker genes in addition to the "FindMarkers" function in Seurat. Marker heatmaps (S1B Fig, Fig 2G) were generated using the genesorteR package version 0.4.2 [47] with a quant setting of

0.97 (S1B Fig) and 0.95 (Fig 2G). Density plots (S3B Fig) were generated using the Nebulosa package version 0.99.94 [48], with standard settings.

**Gene Ontology analysis.** GO networks were generated using ClueGO [49] version 2.5.5 via Cytoscape version 3.7.2 [50]. The network clustering contained data from GO_Biological-Process-EBI-UniProtGOA, REACTOME_ Reactions, REACTOME_Pathways, KEGG_Pathways, WikiPathways, and BioCyc. Clusters contained at least three nodes, and network specificity was adjusted based on the number of genes, which originated from high expressing genes in each cluster analyzed via the "FindMarkers" function in Seurat.

**Gene regulatory network analysis.** Transcription factor regulon activity in specific stromal clusters (S2E Fig) was analyzed using SCENIC version 1.1.2.2 [20], with default settings according to the "running SCENIC" vignette.

**Dataset similarity.** Dataset similarity (S5A Fig) was calculated using matchSCore2 version 0.1.0 [29], with default settings.

**Ligand–receptor interactions.** Ligand–receptor interactions (S3A Fig) were generated using the python package CellphoneDB version 2.1.4 [19], with default parameters.

## Quantification of GFP-positive nuclei

Images acquired on the Leica SP8 confocal microscope were imported into FIJI [51] in the form of Z-stacks and processed by Z-projection (maximum projection). In order to quantify the GFP fluorescence, intensity of individual GFP positive nuclei a threshold (Huang algorithm, auto settings) was applied to the GFP channel. The result was further binarized and improved by filling holes and applying a watershed filter. The nuclei were segmented using the analyze particle function (settings: 8-Infinity, circularity: 0–1). The segmentation was overlaid on the DAPI channel and all nuclei of low quality (e.g., partially sectioned/dim nuclei that were not fully captured in the Z-stack) or of inconclusive position in regards to their crypt-associated position (top, bottom) were excluded. Average fluorescence intensity of the remaining nuclei in the GFP channel was measured and analyzed using Prism 8 (GraphPad, USA).

## Quantitative real-time PCR

For quantitative reverse transcription PCR (RT-qPCR), samples were measured in triplicate and average cycle threshold values were quantified relative to *Hprt* using the ΔΔCT method.

## Supporting information

**S1 Fig. Interrogation of the colonic epithelium reveals a variety of known and unknown growth factors.** (A) Cell numbers of the particular clusters indicated in Fig 1C. (B) Heatmap of genes that are differentially expressed among the epithelial clusters (0.97 quantile), reveals heterogeneity between and similarity within the major epithelial lineages (stem-/transit-amplifying-, secretory-, absorptive-, enteroendocrine cells). (C) Relative expression of epithelial signaling pathway components. (Dot plot, size, and color of the dot represent the percentage of cells which express the transcript and the average expression level within a cluster, respectively). (D) GO enrichment terms for Enterocyte 1. GO, Gene Ontology.
(TIF)

**S2 Fig. Growth factors produced by colonic mesenchymal cells.** (A) GO enrichment terms for SMC. (B) Relative expression of factors involved in WNT, BMP, and RTK/MAPK pathways. (Dot plot, size, and color of the dot represents the percentage of cells which express the transcript and the average expression level within a cluster, respectively). (C) GO enrichment terms for CBFs 1. (D) GO enrichment terms for CBFs 2. (E) Heatmap showing the specific

enrichment for transcription factor regulon activity in specific stromal clusters (scaled regulon activity, SCENIC). Bmp, bone morphogenetic protein; CBF, crypt-bottom fibroblast; GO, Gene Ontology; MAPK, mitogen-activated protein kinase; RTK, receptor tyrosine kinase; SCENIC, single-cell regulatory network inference and clustering; SMC, smooth muscle cell. (TIF)

**S3 Fig. Interactome of colonic mesenchymal cells.** (A) Putative ligand–receptor interactions between CBFs1, CBFs2, CTFs, and colonic epithelial subpopulations. Size of the dots represents the significance of the interaction, and color shows the expression of the ligand and receptor in the interacting cell types (CellphoneDB). (B) Density of *Wnt2*, *Wnt2b*, *Cd81*, *Grem1*, and *Rspo3* expressing cells. (UMAP, color indicates the density of cells expressing transcript). CBF, crypt-bottom fibroblast; CTF, crypt-top fibroblast; UMAP, Uniform Manifold Approximation and Projection. (TIF)

**S4 Fig. Fibroblast identity of CTFs and CBFs.** (A, B) Cryosections of *Pdgfra-H2B-eGFP* mice (A) and *Pdgfra-Cre^ERT2; LSL-tdTomato* mice (B). (A) Vimentin expression (red) in *Pdgfra*⁺ cells (green) on representative *Pdgfra-H2B-eGFP* colonic tissue sections confirms their fibroblast identity. (Scale bar = 40 μm) (1,2) Insets of crypt top and crypt bottom, respectively (Scale bar = 5 μm). (B) *Pdgfra-Cre^ERT2; LSL-tdTomato* lineage tracing (single tamoxifen injection, 1 d.p.i.) reveals morphology of Pdgfra-expressing cells, tdTomato (white) (Scale bar = 10 μm) (C) Flow cytometry analysis for GFP of a mesenchymal single-cell suspension isolated from *Pdgfra-H2B-eGFP* mice. (Raw data: S5 Data) (D, E) Relative expression of *Pdgfra* (D) and colonic niche factors (E) in *Pdgfra*⁺ colonic mesenchymal cells. (UMAP, single cells are colored according to transcript expression). CBF, crypt-bottom fibroblast; CTF, crypt-top fibroblast; d.p.i., days post injection; Pdgfra, platelet-derived growth factor receptor A; UMAP, Uniform Manifold Approximation and Projection. (TIF)

**S5 Fig. Interspecies conservation of colonic mesenchymal subpopulations.** (A) Heatmap showing the similarity (Jaccard index) of murine and human colonic mesenchymal subpopulations (matchSCore2). (TIF)

**S1 Data. This file contains the raw data for the indicated figures.** (A) Transcript expression of *Pdgfra* for Fig 2C. (B) Transcript expression of *Vim*, *Col1a1*, *Acta2*, *Myh11*, *Wnt2*, *Wnt2b*, *Rspo3*, *Grem1*, *Wnt5a*, *Bmp3*, and *Bmp7* for Fig 2D. (C) Numeric values and statistical analysis of the PdgfraEGFP fluorescence intensity quantification in Fig 3C. (D) Raw data and fold change calculation of the qPCR data shown in Fig 3E. (E) Transcript expression of *PDGFRA* for Fig 4C. (F) Transcript expression of *WNT2B*, *WNT5A*, *RSPO3*, *BMP2*, and *GREM1* for Fig 4D. Bmp, bone morphogenetic protein; Pdgfra, platelet-derived growth factor receptor A; qPCR, quantitative polymerase chain reaction. (XLSX)

**S2 Data. FCS file containing the raw data of the FACS sorting of the epithelial cell isolation depicted in Fig 1B.** FACS, fluorescence-activated cell sorting; FCS, flow cytometry standard. (FCS)

**S3 Data. FCS file containing the raw data of the FACS sorting of the mesenchymal cell isolation depicted in Fig 1B.** FACS, fluorescence-activated cell sorting; FCS, flow cytometry standard. (FCS)

**S4 Data. FCS file containing the raw data of the FACS sorting of two populations (low and high) of PdgfraEGFP positive cells depicted in Fig 3D.** FACS, fluorescence-activated cell sorting; FCS, flow cytometry standard.
(FCS)

**S5 Data. FCS file containing the raw data of the FACS sorting for PdgfraEGFP positive cells depicted in S4C Fig.** FACS, fluorescence-activated cell sorting; FCS, flow cytometry standard.
(FCS)

## Acknowledgments

We are grateful to all members of the Basler lab and in particular to J. Reichmuth and J. Schopp for valuable comments and to members of the Moor lab and Schwank lab for advice. For technical help, we thank E. Escher, the Functional Genomics Center Zurich, and the Cytometry Facility of University of Zurich.

## Author Contributions

**Conceptualization:** Michael David Brügger, Tomas Valenta, Hassan Fazilaty, Konrad Basler.

**Data curation:** Michael David Brügger, Hassan Fazilaty.

**Formal analysis:** Michael David Brügger, Hassan Fazilaty.

**Funding acquisition:** Michael David Brügger, Tomas Valenta, George Hausmann, Konrad Basler.

**Investigation:** Michael David Brügger, Tomas Valenta.

**Methodology:** Michael David Brügger, Tomas Valenta.

**Software:** Michael David Brügger, Hassan Fazilaty.

**Supervision:** Tomas Valenta, George Hausmann, Konrad Basler.

**Visualization:** Michael David Brügger, Hassan Fazilaty.

**Writing – original draft:** Michael David Brügger.

**Writing – review & editing:** Michael David Brügger, Tomas Valenta, Hassan Fazilaty, George Hausmann, Konrad Basler.

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
