## [Editor Report · Decision Letter 0]

28 Jul 2020

Dear Dr Valenta, 

Thank you for submitting your manuscript entitled "Distinct populations of crypt-associated fibroblasts act as signaling hubs to control colon homeostasis" for consideration as a Short Report by PLOS Biology. Thank you also for your patience as we completed our editorial process, and please accept my apologies for the delay in providing you with our decision.

Your manuscript has now been evaluated by the PLOS Biology editorial staff as well as by an academic editor with relevant expertise and I am writing to let you know that we would like to send your submission out for external peer review.

Please re-submit your manuscript within two working days, i.e. by Jul 30 2020 11:59PM.

Kind regards,

Ines

--

Ines Alvarez-Garcia, PhD

Senior Editor

PLOS Biology

Carlyle House, Carlyle Road

Cambridge, CB4 3DN

+44 1223–442810

---

## [Decision Letter · Decision Letter 1]

28 Sep 2020

Dear Tomas and Konrad,

Thank you very much for submitting your manuscript "Distinct populations of crypt-associated fibroblasts act as signaling hubs to control colon homeostasis" for consideration as a Short Report at PLOS Biology. Thank you also for your patience as we completed our editorial process, and please accept my apologies again for the delay in providing you with our decision. Your manuscript has been evaluated by the PLOS Biology editors, an Academic Editor with relevant expertise, and by three independent reviewers.

The reviews are attached below. You will see that the reviewers find your results interesting, high quality and a good resource for the field. Thus we are pleased to offer you the opportunity to address all the points raised by the reviewers in a revised version that we anticipate should not take you very long. We will then assess your revised manuscript and your response to the reviewers' comments and we may consult the reviewers again.

We expect to receive your revised manuscript within 1 month.

**IMPORTANT - SUBMITTING YOUR REVISION**

*Resubmission Checklist*

*Published Peer Review*

*PLOS Data Policy*

*Blot and Gel Data Policy*

Best wishes,

Ines

--

Ines Alvarez-Garcia, PhD,

Senior Editor,

ialvarez-garcia@plos.org,

PLOS Biology

Reviewers’ comments

Rev. 1:

In this manuscript, Brügger et al. characterize the epithelial and mesenchymal/fibroblast populations of the mouse colon using sc-RNAseq. They compare this to the human colon cell populations and find they are quite comparable. The story is descriptive (which I do not intend to sound like a bad thing) and results in the identification of three different populations of fibroblasts: two associated with the bottom of the crypt (CBF1 and CBF2) and one associated with the top of the crypt (CTF). These populations express different signaling factors that either promote stemness (CBF: Wnt2, Wnt2b and Rspo3) or differentiation (CTF: Wnt5a and Bmps). The authors also provide evidence that these populations correspond to spatially separated Pdgrfa+ populations in situ (Pdgrfa-high: CTF, low: Pdgrfa-CBF1 and 2).

The manuscript is clearly written, with a balanced and focused introduction. The dataset (>7000 cells total, >3000 transcripts per cell) has been deposited at NCBI GEO and promises to be a useful resource for scientists interested in the interaction between intestinal stem cells and their presumed niche.

This is a straightforward study and I only have a few remarks that perhaps the authors can consider to clarify a few points:

1. The introduction mentions that most other studies use specific cell isolation protocols. It is not clear what the authors refer to, given that they themselves of course also use an enzymatic digestion and cell isolation protocol (i.e. FACS sort for EPCAM+ and EPCAM- cells). I think I get what they mean (theirs is 'just' a sort for epithelial versus non-epithelial), rather than a specific enrichment for a subpopulation, but perhaps they can clarify this a bit.

2. In Figure 1B, does this refer to the input prior to sorting or is this purity after sorting?

3. Related to the sorting: Can the authors comment on and/or quantify the ratio of these different cell populations (CBF1/2 vs CTF vs epithelial cells)?

4. Figure 2 and throughout: Can the authors speculate on the difference between the CBF1 and CBF2 population? Are there any discriminatory marks that separate these two otherwise functionally overlapping populations? Based on 2G this seems a logical step to include in this manuscript and something the authors should be able to do with the tools and analysis pipeline at hand, or is this something the authors plan to follow up on later? Alternatively, could it be some sort of experimental artifact? A list of differentially expressed genes in these populations might be informative.

5. From the legend of Figure 3A-C it is not clear that we are looking at cryosections, but I imagine that this is what is depicted, based on the methods?

6. Related to 5: Has the GFP signal been corrected for DAPI? The authors describe in the methods how they performed the GFP quantification, but this seems to related to masking of nuclei to include. Ideally, the GFP signal is corrected for the DAPI intensity to make up for depth/sectioning/processing/imaging differences at different locations on the slide. If it is not done or cannot be done, this should be noted in the legends and/or methods.

Rev. 2: Shalev Itzkovitz – this reviewer has waived anonymity

In this work, Brugger et al. perform single cell RNAseq experiments to characterize the mesenchymal cell types constituting the colonic crypt niche. They uncover three distinct Pdgfra+ fibroblast populations, expressing distinct antagonistic morphogens. More specifically, crypt bottom fibroblasts, which express canonical Wnt ligands, Rspo3 and Bmp inhibitors, and crypt top fibroblasts, expressing non-canonical Wnt ligands and Bmp ligands. The study is well performed and will be an important resource for biologists interested in stem cell biology. The following points should be addressed in a revision:

- The supplementary figures nicely show the expression of ligands of major signaling pathways among the three fibroblast populations, however the paper would be strengthened by a more comprehensive analysis of ligand-receptor interactions, most importantly between the fibroblasts and epithelial compartments. Specifically, what are the main elevated epithelial ligands expressed in tip/crypt enterocytes with matching elevated tip/crypt fibroblast receptors and vice versa? There are several examples, e.g. Pdgfra in the crypt tip enterocytes and Pdgfra in the CTF, Rspo3 in CBF and matching epithelial receptor Lgr5 at the crypt bottom. Such analysis can be performed by tools such as https://www.cellphonedb.org/ or NicheNet (https://www.nature.com/articles/s41592-019-0667-5). Similarly, are there interactions between the fibroblast populations and pericytes/endothelial clusters?

- Can the authors comment about the potential spatial differences between CBF1 and CBF2? Is one population closer to the epithelium? Do any of these populations overlap trophocytes (https://www.nature.com/articles/s41556-020-0567-z)?

- Do CTFs express Lgr5, as has been shown for villus tip telocytes (VTTs, ref 20)? The authors should comment on the overlap and differences in gene expression between CTFs and VTTs, which express many of the discussed CTF markers, such as the non-canonical Wnt5a, Bmp ligands and F3.

- It would be informative to add immunofluorescence for PDGFRA protein, which labels the entire cell bodies. Telocytes have a unique morphology with elongated processes that 'wrap around' the crypts, as well as tiny 'telopode' protrusions into the lamina propia, such staining for PDGFRA would reveal such morphologies in the colon.

- The expression of both the canonical Wnt ligands and antagonists is intriguing, do the same CBF cells express both ligands and antagonists, or are they expressed in different CBF subsets?

- Row 213 - add ref for the connection between the Lef1 regulon and Lgr5 expression.

- Row 406 - add ref. for "McCarthy and colleagues"

Rev. 3:

Summary of opinion: The instructions to the reviewer start "Short Reports should be novel, provocative and of general interest". The findings here are of general interest and high quality. They are not particularly novel, and because they support and extend prior studies, they are only modestly provocative. Overall, this manuscript contributes to the research field of scRNAseq and stem cell regulation, revealing the heterogeneous cell populations that regulate cellular homeostasis in the colon. Given the useful and high quality confirmatory, rather than novel, nature of the manuscript, I defer to the editor to decide if it meets the journal's editorial guidelines.

General comments: Brügger et al. aim to identify mesenchymal cell populations regulating proliferation and differentiation of colonic stem cells located in the crypt base. To look into the heterogenous nature of the intestinal mesenchymal tissue, scRNAseq was performed. The authors identified three distinct PDGFRA positive fibroblast subpopulations expressing Wnt or Bmp signals that regulate stem cell fate.

Generally, the setup of experiments is comprehensible and the authors present their data in a logic and conclusive way. The experiments and the interpretation of the data appear reliable and solid. The cell number (7395) and the read depth (3243/cell) is robust and will be a useful resource.

However, in a recent study published by McCarthy and colleagues (and cited appropriately) nearly identical results were presented. Here, a very similar experimental setup was used to identify mesenchymal cell populations regulating stem cells in the small intestine. McCarthy et al found the same cell populations with very similar gene expression profiles and localizations within the intestinal tissue. Therefore, this study by Brüegger et al has largely a confirmative character, since it reflects and verifies the results of the study from McCarthy and colleagues and doesn't break much new ground.

Additionally, by reanalysis of a recently published dataset, Brüegger et al could show that similar cell populations exist in human colonic tissue what might make this study useful for researchers focused on tumor initiation and progression.

Overall, this manuscript contributes to the research field of stem cell regulation revealing the heterogeneous cell populations which fulfil the regulation of the sensitive cellular homeostasis in the colon.

Major points:

CBF1 and CBF2 were shown to represent 2 distinct cell populations. However, the authors did not show data about the spatial localization of both cell types within the tissue.

Are both CBF1 and 2 needed to support stem cell growth? Or can this be achieved by only CBF1 or CBF2? Organoids would be a suitable model to examine this issue. According to the title, CBFs act as hubs to control colon homeostasis, what is not reflected in the experimental results.

Line 401: The statement "Canonical Wnt ligands, which are integral to intestinal epithelial stem cell maintenance, are secreted by epithelial Paneth cells in the small intestine" is misleading and should be corrected. Multiple studies from multiple labs have shown that Paneth cells, Wnt3 in intestinal epithelium, and Porcn and Wls in the intestinal epithelium are all completely dispensable for normal small intestine function.

Minor points:

Page 5, line 100: non-epithelial (EpCAM-, CD45+) cells should be ***CD45-*** cells

Page 6, line 132: Dll1 and Dll4 expression is displayed in S1D not S1C

Page 6, line 135: GO-terms are shown in S1C not S1D

Page 6, line 137" Egf family ligands and ephrins expression can be found in S1D not S1C

Page 7, line 139: Ihh expression is shown in S1D not S1C

Sometimes the reference to the figures is missing in the text, what makes it difficult for the reader to follow the results, e.g.

- page 6, line 127: Lgr5, Olfm4, Axin2 and Mki67. Moreover, Olfm4 and Mki67 expression is not displayed in the figures

- page 6, line 133: (Guaca2a+, Alpi+, Aldh1/1+)

Page 8, line 183: S1 and S1 should be S2

Page 10, line 233: Expression of Vimentin cannot be found in Fig. S3A

---

## [Editor Report · Decision Letter 2]

13 Nov 2020

Dear Tomas and Konrad,

Thank you for submitting your revised Short Reports entitled "Distinct populations of crypt-associated fibroblasts act as signaling hubs to control colon homeostasis" for publication in PLOS Biology. I have now discussed your revision with the team of editors and obtained advice from the Academic Editor. 

We're delighted to let you know that we're now editorially satisfied with your manuscript. However before we can formally accept your paper and consider it "in press", we also need to ensure that your article conforms to our guidelines. A member of our team will be in touch shortly with a set of requests. As we can't proceed until these requirements are met, your swift response will help prevent delays to publication. Please also make sure to address the data and other policy-related requests noted at the end of this email.

- a cover letter that should detail your responses to any editorial requests, if applicable

*Copyediting*

*Published Peer Review History*

*Early Version*

Best wishes,

Ines

--

Ines Alvarez-Garcia, PhD

Senior Editor,

PLOS Biology

ETHICS STATEMENT:

-- Thank you for sending us the details of the ethics permit. Please include the use protocol/permit/project license.

Fig. 1B; Fig. 2C, D; Fig. 3D, E; Fig. 4C, D and Fig. S4C

Also, please make available the data that you have deposited in GEO with the accession number GSE151257

---

## [Editor Report · Decision Letter 3]

25 Nov 2020

Dear Dr Valenta,

On behalf of my colleagues and the Academic Editor, Emma Rawlins, I am pleased to inform you that we will be delighted to publish your Short Reports in PLOS Biology. 

PRODUCTION PROCESS

Before publication you will see the copyedited word document (within 5 business days) and a PDF proof shortly after that. The copyeditor will be in touch shortly before sending you the copyedited Word document. We will make some revisions at copyediting stage to conform to our general style, and for clarification. When you receive this version you should check and revise it very carefully, including figures, tables, references, and supporting information, because corrections at the next stage (proofs) will be strictly limited to (1) errors in author names or affiliations, (2) errors of scientific fact that would cause misunderstandings to readers, and (3) printer's (introduced) errors. Please return the copyedited file within 2 business days in order to ensure timely delivery of the PDF proof. 

If you are likely to be away when either this document or the proof is sent, please ensure we have contact information of a second person, as we will need you to respond quickly at each point. Given the disruptions resulting from the ongoing COVID-19 pandemic, there may be delays in the production process. We apologise in advance for any inconvenience caused and will do our best to minimize impact as far as possible.

EARLY VERSION

PRESS 

Kind regards,

Erin O'Loughlin

Publishing Editor, 

PLOS Biology

on behalf of

Ines Alvarez-Garcia,

Senior Editor

PLOS Biology